# The Repellent Capacity against *Sitophilus zeamais* (Coleoptera: Curculionidae) and *In Vitro* Inhibition of the Acetylcholinesterase Enzyme of 11 Essential Oils from Six Plants of the Caribbean Region of Colombia

**DOI:** 10.3390/molecules29081753

**Published:** 2024-04-12

**Authors:** Amner Muñoz-Acevedo, María C. González, Jesús E. Alonso, Karen C. Flórez

**Affiliations:** 1Department of Chemistry and Biology, Universidad del Norte, Puerto Colombia 081007, Colombia; mgonzalesd@uninorte.edu.co; 2Department of Mathematics and Statistics, Universidad del Norte, Puerto Colombia 081007, Colombia; jcabrera@uninorte.edu.co (J.E.A.); lozanok@uninorte.edu.co (K.C.F.)

**Keywords:** repellency, AChE inhibition, maize weevil, Colombian essential oils, post-harvest protection, maize

## Abstract

The repellent capacity against *Sitophilus zeamais* and the *in vitro* inhibition on AChE of 11 essential oils, isolated from six plants of the northern region of Colombia, were assessed using a modified tunnel-type device and the Ellman colorimetric method, respectively. The results were as follows: (i) the degree of repellency (DR) of the EOs against *S. zeamais* was 20–68% (2 h) and 28–74% (4 h); (ii) the IC_50_ values on AChE were 5–36 µg/mL; likewise, the %inh. on AChE (1 µg/cm^3^ per EO) did not show any effect in 91% of the EO tested; (iii) six EOs (*Bursera graveolens*—bark, *B. graveolens*—leaves, *B. simaruba*—bark, *Peperomia pellucida*—leaves, *Piper holtonii* (1b*)—leaves, and *P. reticulatum*—leaves) exhibited a DR (53–74%) ≥ C^+^ (chlorpyrifos—61%), while all EOs were less active (8–60-fold) on AChE compared to chlorpyrifos (IC_50_ of 0.59 µg/mL). Based on the ANOVA/linear regression and multivariate analysis of data, some differences/similarities could be established, as well as identifying the most active EOs (five: *B. simaruba*—bark, *Pep. Pellucida*—leaves, *P. holtonii* (1b*)—leaves, *B. graveolens*—bark, and *B. graveolens*—leaves). Finally, these EOs were constituted by spathulenol (24%)/β-selinene (18%)/caryophyllene oxide (10%)—*B. simaruba*; carotol (44%)/dillapiole (21%)—*Pep. pellucida*; dillapiole (81% confirmed by ^1^H-/^13^C-NMR)—*P. holtonii*; mint furanone derivative (14%)/mint furanone (14%)—*B. graveolens*—bark; limonene (17%)/carvone (10%)—*B. graveolens*—leaves.

## 1. Introduction

Maize (*Zea mays* L.) is one of the oldest, best-known, and most essential grasses from America, in addition to being second of the most produced cereals in the world (ca. 197 million ha cultivated; annual production > 1 billion metric tons; yield: 6 tons/ha) [1,2]; its consumption is massive because it is included as part of the primary food group of the family basket (at any socioeconomic level) due to its low cost, mainly in some countries belonging to sub-Saharan Africa, Latin America, and Asia, where the major production areas are also located [1,2,3,4]. As maize is a versatile multipurpose crop, besides its usage as a staple food, it has cultural/environmental, nutritional, and economic impacts, as well as feed/forage, energy, and industrial uses [2,5,6,7]. In this way, one of the emerging uses of corn is in animal feed, which has accelerated and boosted the demand for maize, e.g., in Asia [2,8]. Therefore, corn is playing a dynamic role in the worldwide agri-food systems and food security (2030 Agenda for Sustainable Development) [9,10,11,12].

When considering Latin America, which is the center of origin/diversity of maize (ca. 194 native populations, 131 distinctive races, and 23 countries), ca. 30 million tons of grain per year (on 10 million ha) are produced mainly by Argentina, Mexico, and Brazil; the last two countries are the leading producers in South America, but Mexico is the one that produces most of the maize for direct human consumption. Then, from the diversity of maize, many food products and recipes are manufactured for human nutrition [13,14,15]. In the case of Colombia, maize is the third most important crop (largest planted area ca. 13%) of agricultural production and is one of the most relevant crops in the agri-food sector; it is a crop of small producers (~60% up to 10 ha). In addition, Colombia is the first importer of this cereal in South America and the seventh in the world; in 2016, 74% of the national demand was imported. According to projections for 2030, an increase in demand would be expected, which will require an increase in imports by 39%. However, a proposed strategy will respond to the challenge of producing more and better. This strategy is based on planting ca. 1,300,000 ha of technified maize with improved seeds, sustainable agriculture, irrigation, and conservation adapted to climate change. That would allow an average yield of ~6 tons/ha of maize to be achieved, thereby completely reducing grain imports and guaranteeing the food security of the Colombian people [16,17,18].

Despite this, there is worldwide trouble that affects cereal/grain producers, and it is the loss of grain (ca. 30–40% of total production in Latin America) during post-harvest storage due to various factors (e.g., pests, inadequate storage conditions), which influence its quality [19]. One of the most persistent, harmful, and primary pests is *Sitophilus zeamais* Motschulsky (the maize weevil), which infests and attacks stored maize grains (leaving them in total deterioration) in tropical/subtropical regions [20]. Furthermore, this insect can damage other cereals/grains such as sorghum, wheat, rice, and certain industrialized dry products [19,21]. To exterminate/control this type of pest, some alternatives have been used for the integrated management, biological (e.g., the introduction of natural enemies), physical (e.g., manipulation of temperature and relative humidity conditions), and chemical (e.g., pesticides (organophosphates/organochlorines/carbamates)), each with their advantages/disadvantages [19,22].

Exposure to pesticides has been one of the most effective methods due to the mechanism of action involved (inhibition of the acetylcholinesterase enzyme (AChE) [23]). One of them is chlorpyrifos (CP), which is a well-known, practical, and common organophosphate pesticide widely used in households and agriculture (for protection of crops against insects, e.g., corn [24]). It has a semi-volatile chemical nature (P_vap_. 0.0025 Pa; KH: 6.6 × 10^−6^ atm-m^3^/mol (KH < 10^−5^ atm-m^3^/mol can volatilize slowly) and onion/garlic or slightly skunky/mild mercaptan odor); its volatilization is a significant dissipative process in the environment, and in addition, it is one of the few pesticides that has shown moderate toxicity through inhalation (>0.2 mg/L in rats, between 4 and 6 h [25]). However, its use entails significant disadvantages (for all pesticides), such as the development of resistance (by insects), environmental pollution (as effluents), lack of selectivity (damage to other organisms, including humans), and high toxicity (causing death), among others [26,27].

Since the advent of green chemistry/toxicology and sustainable development [28], an environmentally friendly, low-cost, low-risk-to-humans, naturally occurring, and renewable alternative called “biopesticides” or “biochemical pesticides” has emerged [29]. This type of substance includes various extracts, essential oils (EOs), and compounds isolated from plants [30,31,32,33]. In addition, some authors [34,35,36] reported the ability of certain plants and their components, e.g., alkaloids (e.g., physostigmine and galantamine) and terpenoids (e.g., ascaridol, carvacrol, *p*-cymene, elemol, α-pinene, and viridiflorol, among others [37]), to inhibit AChE. The last type of compound is the main constituent of EOs.

On the other hand, Colombia is one of the six so-called megadiverse countries, with ca. 10% of the world’s biodiversity distributed between natural forests (e.g., tropical dry forest) and savanna and wetland areas in its continental portion. Based on Rangel´s report [38], there are 20300–26500 ± 1000 species of angiosperms (several in the wild), some of which have been used as food and in ethnomedicine by communities (e.g., peasant, Afro-descendant, and Indigenous farmers). However, there is no information on its possible application for other species. The species *Bursera graveolens*, *B. simaruba*, *Peperomia pellucida*, *Piper haugtii*, *P. holtonii*, and *P. reticulatum* are plants found in the tropical dry forest of the northern Colombian region (Departamentos de Atlántico/Sucre) and two of these plants (*B. graveolens* and *B. simaruba*) have been traditionally applied as insecticides and mosquito repellents [39,40,41].

This work aimed to establish the degree of repellency against the maize weevil (*Sitophilus zeamais* (Coleoptera: Curculionidae)) and the *in vitro* inhibitory capacity on AChE of 11 EOs isolated from six plants (*B. graveolens*, *B. simaruba*, *Pep. pellucida*, *P. haugtii*, *P. holtonii*, and *P. reticulatum*) of the tropical dry forest located in the northern region of Colombia, using a modified tunnel-type device and the Ellman colorimetric method, respectively. Likewise, CP was chosen/used as a positive control for the research due to all the characteristics previously described: (i) wide use in agriculture (e.g., corn crop), (ii) characteristic odor (allowing its application as a fumigant related to “odor” and “volatility”), and (iii) moderate toxicity through inhalation. In addition, all data were statistically treated using an ANOVA (two-way and one-way combined with linear regression) and multivariate analysis to find some criteria for differentiation/similarity and significance. Finally, the chemical compositions of the EOs were determined using GC-MS according to all the corresponding rigorous criteria.

## 2. Results

### 2.1. Identity of the Plants

The six botanical samples were identified as *Piper holtonii* C. DC. (No. COL 578342), *P. haugtii* (No. COL 579231), *P. reticulatum* L. (No. COL 589613), *Peperomia pellucida* (No. COL 578363), *Bursera graveolens* (No. COL 560956), and *B. simaruba* (No. COL 574668).

### 2.2. Chemical Composition of the Essential Oils

Table 1 presents the chemical composition (the most abundant constituents) determined using GC-MS of each EO isolated according to the collection location and part of the plant used. Thus, the most important constituents of EOs were dillapiole, carotol, spathulenol, limonene, mintlactone/its derivative, caryophyllene oxide, β-elemene, and β-pinene.

A complementary chemical analysis using ^1^H-/^13^C-NMR (Appendix A) was performed on the EO with the highest content of dillapiole (*P. holtonii*—1bL*) to verify the structure of the constituent. The assignment of structure-related signals is presented below. Leaf EO: colorless liquid denser than water. Dillapiole (81%, main constituent)—C_12_H_14_O_4_. GC-MS (EI, 70 eV), *m*/*z* (%): 222.05 (M^+•^, 100). ^1^H-NMR (CDCl_3_, 400 MHz): δ 6.35 (1H_Ar_, “s”), 5.96–5.86 (1H, “m”, -CH=C), 5.88 (2H, “s”, -O-CH_2_-O-), 5.07–5.04 (1H_cis_, “m”, -C=CH_2cis,trans_), 5.03–5.02 (1H_trans_, “m”, -C=CH_2cis,trans_), 4.01 (3H, “s”, -O-CH_3_), 3.75 (3H, “s”, -O-CH_3_), 3.31–3.29 (2H, “m”, -CH_2_-C=) ppm. ^13^C-NMR (CDCl_3_, 100 MHz): δ 144.7 (-C_Ar_-), 144.4 (-C_Ar_-), 137.7 (-C_Ar_-), 137.5 (-CH=), 136.0 (-C_Ar_-), 126.2 (-C_Ar_-), 115.7 (=CH_2_), 108.9 (-CH_Ar_-), 101.2 (-O-CH_2_-O-), 61.4 (-O-CH_3_), 60.1 (-O-CH_3_), 34.0 (-CH_2_-) ppm. The signals, multiplicities, and couplings for each aromatic/olefinic/methyl/methylene H and C in the NMR spectra of the EO are matched with the same signals, multiplicities, and couplings in the NMR spectra for dillapiole, as reported by Cicció and Ballesteros [42] and Rojas et al. [43].

### 2.3. Degree of Repellency against Maize Weevils by Essential Oils

The results of the repellent effect of the 11 EOs against *S. zeamais* were recorded and are displayed in Table 2 and Appendix A, respectively. At the exposure times tested, the repellency percentages were 20 ± 0%–68 ± 8% (relative standard deviation—%RSD: 0–17%) and 28 ± 4%–74 ± 11% (%RSD: 9–18%) at 2 h and 4 h, one-to-one. Thus, considering the effect for each hour (Appendix A), the EOs 1bL* (*P. holtonii*), 2aL (*Pep. pellucida*), 5aB (*B. graveolens*), and 6aB (*B. simaruba*) presented equal or higher repellency than chlorpyrifos (C^+^) at 2 h of the experiment. In contrast, the effect of EOs 1bL* (*P. holtonii*), 2aL (*Pep. pellucida*), and 6aB (*B. simaruba*) remained higher than that of C^+^ at 4 h. In this last hour, the EOs 4bL^‡^ (*P. reticulatum*) and 5aL (*B. graveolens*) increased the effect to values higher than chlorpyrifos.

The first approach for the statistical treatment of data was via ANOVA with two variables (exposure time and type of EO), which allowed the proposal of the two-way fixed-effect model (1) (Appendix A). For this model, the assumptions of normality of residuals (K-S, *p*: 0.078), homogeneity of variances (Levene´s test *p*: 0.211), and independence of errors were verified. The model proved to be significant, as observed in the ANOVA in Appendix A. Nonetheless, only the effect of the “type of EO” (*p* < 0.00001) along with the interaction between the “type of EO” and “exposure time” (*p* < 0.00001) were significant, while the exposure time as a main effect was not significant (*p*:0.075); consequently, the resulting model (1) was simplified/reduced due to the last assumption. The comparison between each EO (11) with the positive control (C^+^) using Dunnett´s test revealed that five EOs (1bL* (*P. holtonii*), 2aL (*Pep. pellucida*), 5aL (*B. graveolens*—L), 5aB (*B. graveolens*—B), and 6aB (*B. simaruba*)) did not present statistically significant differences (significance level of 5%) with C^+^ (*p* > 0.05); these EOs had similar repellency percentages.

Since model (1) excluded the effect of exposure time, a second approach for data treatment was necessary; thereby, model (2) was proposed (Appendix A). After verifying the assumptions, it was found that the errors showed a normal distribution (K-S, *p*: 0.078), and the variances were constant at each level (Levene´s test, *p*: 0.18). The results of the test of effects and regression coefficients were significant (*p* < 0.00001), as was model (2) (Appendix A).

Concerning the results listed in Appendix A and plotted in Figure 1, it could be determined that three EOs (4bL^†^ (*P. reticulatum*), 5aB (*B. graveolens*), and 6aL (*B. simaruba*)) decreased the degrees of repellency on maize weevils over time; in contrast, three more EOs (4bL^‡^ (*P. reticulatum*), 5aL (*B. graveolens*), and 6aB (*B. simaruba*)) increased the repellent capacity, surpassing C^+^. Otherwise, the degree of repellency of two other EOs (1bL* (*P. holtonii*) and 2aL (*Pep. pellucida*)) was statistically like that of chlorpyrifos and was not dependent on the exposure time. Meanwhile, EO 3bL (*P. haugtii*) increased its repellency percentage over time but did not surpass C^+^; it also had the lowest effect. As a final point, EOs 1bL** (*P. holtonii*) and 1aL (*P. holtonii*) presented a negative repellency effect (<<C^+^) independent of exposure time.

### 2.4. Inhibitory Effect on AChE Enzyme by Essential Oils

The *in vitro* AChE inhibition assay was carried out to establish the degree of toxicity of the EOs; the results are presented in Appendix A and Figure 2. Referring to the table, all EOs showed IC_50_ values ranging between 4.8 ± 0.6 µg/mL and 36 ± 3 µg/mL, while 0.59 ± 0.04 µg/mL was the IC_50_ value for the positive control. Likewise, when each EO was prepared at 1 µg/mL to determine its percentage inhibition on the AChE enzyme, only one of them showed a quantitative result (10.4 ± 0.5%—6aL (*B. simaruba*)); for the remaining EOs, no value (inhibition values less than zero) could be determined, which would allow us to infer that the samples were not active under these conditions. The percentage inhibition value of chlorpyrifos at 1 µg/mL was 59 ± 3%. It is worth noting that none of the EOs tested overcame the inhibitory effects (both IC_50_ and %I) of the positive control.

Even so, the most active EO was 6aL (*B. simaruba*), which could evidence some inhibitory effect on the AChE enzyme, that is, the lowest IC_50_ value (4.6 ± 0.3 µg/mL) and the highest %I value (10.4 ± 0.5%) with respect to the other EOs (Figure 2). Comparison between C^+^ and EO 6aL (*B. simaruba*) showed that chlorpyrifos was ca. eight and six times more active than 6aL, based on the IC_50_ and %I values, respectively.

### 2.5. Multivariate Statistical Analysis Applied to the Results of Biological Tests of EOs

Once the data obtained (from the biological assays carried out on the EO samples) were statistically treated, these data were also subjected to a multivariate analysis (principal component analysis—PCA, cluster analysis—CA, and K-means clustering analysis—KmCA) to find any relationship between the bio-tests and the 11 EOs. In this manner, the MVA was initiated by applying PCA, in which Factor 1 (~55%) and Factor 2 (~27%) together could explain ca. 82% of the variability of the original dataset; these factors presented eigenvalues higher than or equal to 1.0 (F1—~2.2; F2—~1.1). In addition, the factorability of the variables was examined using the measure of sampling adequacy, considering the Kaiser–Meyer–Olkin (KMO) parameter and Bartlett´s test of sphericity. The KMO value was 0.582043 (must be >0.5), and Bartlett´s test was significant (*p*: 0.107015; *p* > 0.05), indicating that there was a certain degree of collinearity between the variables and that the identified factors were consistent.

Figure 3 shows the resulting PCA graph, and according to it, the variables with the most significant contribution (based on correlation) to Factor 1 were the values of IC_50_ (0.335850) and repellency at 2 h (0.315752). In comparison, the values of %I (0.482894) and repellency at 4 h (0.343194) contributed mainly to Factor 2 (e.g., the fitted equation to the first principal component was 0.561918 × 2 h + 0.439912 × 4 h − 0.579526 × IC_50_ + 0.393542 × %I (1 ppm), which clearly shows the highest contribution by the IC_50_ value and exposure time (at 2 h)). In another way, the cases with the most significant contribution (based on correlations) to Factor 1 were mainly C^+^ (30.94731) and EOs 3bL (30.06854), 1aL (15.05037), and 6aB (8.04253), while C^+^ (37.99287) and EOs 6aL (17.63257), 1bL* (16.34247), and 2aL (7.60727) contributed notably to Factor 2. As a conclusive observation, in agreement with the PCA graph, three main groups were found based on the similarities between the biological results: I—including C^+^ (chlorpyrifos); II—composed of six EOs (1bL** (*P. holtonii*), 1aL (*P. holtonii*), 3bL (*P. haugtii*), 4bL^†^ (*P. reticulatum*), 4bL^‡^ (*P. reticulatum*), and 6aL (*B. simaruba*)); III—made up of five EOs (1bL* (*P. holtonii*), 2aL (*Pep. pellucida*), 5aL (*B. graveolens*—L), 5aB (*B. graveolens*—B), and 6aB (*B. simaruba*)).

For its part, regarding the CA using the single linkage as an amalgamation (joining) rule and the Euclidean distances (non-standardized) as a linkage measure, the vertical hierarchical tree plot was depicted, including the 11 EOs and chlorpyrifos along with their biological results. Under the similar characteristics from the 12 cases, three clusters were established (Figure 4): I—chlorpyrifos (C^+^); II—constituted by five EOs (1bL** (*P. holtonii*), 1aL (*P. holtonii*), 3bL (*P. haugtii*), 4bL^†^ (*P. reticulatum*), and 6aL (*B. simaruba*—L)); III—composed of six EOs (1bL* (*P. holtonii*), 2aL (*Pep. pellucida*), 4bL^‡^ (*P. reticulatum*), 5aL (*B. graveolens*—L), 5aB (*B. graveolens*—B), and 6aB (*B. simaruba*—B)). In addition, some sub-clusters were observed with the lowest distances between the cases of clusters II and III: i.—2aL and 6aB (8.0), ii.—4bL^‡^ and 5aL (11.4), iii.—1bL** and 1aL (17.2), iv.—4bL^†^ and 6aL (20.6). The comparison between groups II and III obtained using PCA and CA showed a single difference associated with the location of EO 4bL^‡^ (*P. reticulatum*).

The same cluster number (three) as in CA was pre-established for the KmCA. Then, the analysis of variance applied to the KmCA, as a differentiation criterion, showed that the variables %I (*p*: 0.00000) and exposure time (at 4 h) (*p*: 0.00021) were significant (*p* < 0.05). Figure 5 displays the graph of the means for the three clusters according to the values of IC_50_, %I (at 1 µg/mL), and degree of repellency (at 2 h and 4 h). Thus, cluster 1 consisted of C^+^; cluster 2 contained the EOs 1bL* (*P. holtonii*), 2aL (*Pep. pellucida*), 4bL^‡^ (*P. reticulatum*), 5aL (*B. graveolens*), 5aB (*B. graveolens*), and 6aB (*B. simaruba*). In contrast, cluster 3 was made up of the EOs 1bL** (*P. holtonii*), 1aL (*P. holtonii*), 3bL (*P. haugtii*), 4bL^†^ (*P. reticulatum*), and 6aL (*B. simaruba*). To close the interpretation of K-means, the EOs that presented the expected behavior, based on the degree of repellency (≥C^+^) and AChE inhibition (IC_50_ >>> C^+^, I% (1 ppm) <<< C^+^), were in cluster 2. Among these six EOs, five of them (1bL* (*P. holtonii*), 2aL (*Pep. pellucida*), 5aL (*B. graveolens*—L), 5aB (*B. graveolens*—B), and 6aB (*B. simaruba*—B)) were the most active.

## 3. Discussion

As a starting point, the comparison of the chemical compositions of the 11 EOs with the scientific literature consulted showed some crucial differences; for instance, the EOs of the *P. holtonii* leaves under study were composed of dillapiole (64–81%) while the EO of the species reported by Pineda et al. [44] was constituted by apiol (64%, a structural isomer of dillapiole). In addition, EOs isolated from leaves/aerial parts from the Brazilian *Pep. pellucida* were composed of dillapiole (40–55%)/(*E*)-caryophyllene (11–14%) [45,46] and dillapiole (37%)/carotol (13%) [47], while the EO of the Colombian species consisted of carotol (44%)/dillapiole (21%); nevertheless, this chemical composition presented similarity in the main constituents and some differences in the relative quantities (carotol (32%)/dillapiole (21%)) with the EO of the Indian species [48]. Likewise, EOs of leaves from Panamanian, Brazilian, and Peruvian *P. reticulatum* contained β-selinene (19%)/β-elemene/α-selinene (16% for each) [49], β-elemene (25%)/β-caryophyllene (17%) [50], and apiol (15%)/germacrene D (13%) [51], respectively, in contrast to the EOs from the Colombian shrub whose main compounds were β-pinene (8%)/β-elemene (8–9%)/germacrene D (8%).

Considering *Bursera* species, the EOs of leaves/aerial parts from Cuban/Mexican *B. graveolens* were constituted by limonene (26–43%), (*E*)-β-ocimene (13–21%), and β-elemene (11–14%) [52,53,54]; stem/branch EOs from Ecuadorian trees contained limonene (35–59%)/α-terpineol (11–13%) [55,56] or viridiflorol (71%) [57], while the trunk EOs from Peruvian species were represented by limonene (77%), limonene (14%)/α-terpineol (13%), or α-terpinene (32%) [58,59,60]. The EOs isolated from leaves/stem from Colombian trees contained limonene (42%)/pulegone (21%)—leaves [61], or limonene (48%)/caryophyllene oxide (14%)—leaves, and limonene (42%)/menthofuran (15%)—stem [62], whose compositions differed from those of this work, which were limonene (17%)/carvone (10%)—leaves and mintlactone and its derivative (14% for each one)—bark.

In regards to *B. simaruba*, the EOs from Jamaican species were constituted by α-pinene/(*E*)-cadina-1(6),4-diene (10% for each)/β-caryophyllene (9%)—leaves, and α-pinene (32%)/β-pinene (14%)—bark [63]; the EOs from Costa Rican trees consisted of o-cymene (65%)—leaves, and α-phellandrene (29%)/β-caryophyllene (19%)/o-cymene (13%)/α-thujene (12%)—bark [64]; in comparison, the leaf EO from Guadeloupe presented limonene (47%)/β-caryophyllene (15%)/α-humulene (13%) [65]. In contrast, the EOs from the Colombian tree contained caryophyllene oxide (12%)/spathulenol (11%)—leaves and spathulenol (24%)/β-selinene (18%)/caryophyllene oxide (10%)—bark; nevertheless, the previous composition showed differences for the branch EO (caryophyllene oxide (18%)/β-caryophyllene (10%)) from the Venezuelan tree [66]. Lastly, the chemical composition of the leaf EO from *P. haughtii* is reported for the first time.

On the other hand, it is well known that plant EOs have shown both contact-fumigant toxicities/repellency against insects (e.g., Coleoptera, Diptera, etc.) [67,68,69,70,71,72,73,74] and *in vitro* inhibition of AChE [75,76,77,78,79,80,81]. Then, when the main plant species (containing EOs) with the highest repellent/fumigant capacities against Coleoptera were reviewed, these belonged to the genera *Cymbopogon*, *Ocimum*, and *Eucalyptus*, whose chemical compositions were mainly based on monoterpenoids, e.g., neral/geranial, nerol/geraniol, citronellol/citronellal, eucalyptol or limonene/pinenes [74,82,83,84,85,86,87,88,89]. The EOs containing these monoterpenoids presented a high degree of repellency (%R ca. 42–96%, at 0.016–16 nL/cm^2^, 0.15–0.5%, or 16–84 µL/L) against *Tribolium castaneum* (red flour beetle), which is considered one of the two most common secondary pests of all stored gramineous/cereal products worldwide, according to the “post-harvest compendium” of FAO of the UN [90]. In addition, other terpenoids, e.g., phenolic monoterpenoids (thymol/carvacrol), sesquiterpenoids (β-caryophyllene, caryophyllene oxide), or phenylpropa(e)noids, have also demonstrated repellent/fumigant capabilities against Coleoptera [91,92,93,94,95,96].

If the plants under study are considered, the EOs of most of these plants did not involve any report related to the two biological models tested, except for *P. holtonii*. Even so, some reports [61,97,98,99,100,101,102,103,104] related to repellent/fumigant activities of EOs from Colombian plants (wild/domesticated/local market—from Arauca, Bogotá/Cundinamarca, Bolivar, Boyacá, Meta, Tolima, Santander) against Coleoptera (*T. castaneum* or *S. zeamais*) were found in the reviewed scientific literature, and three reports [61,97,98] dealt with the repellent capacity of EOs from Colombian *B. graveolens* (leaves; northern region) and *P. holtonii* (aerial parts) against *T. castaneum* and *S. zeamais*, respectively. The other plants were *Artemisia dracunculus* (estragol), *Cananga odorata* (benzyl acetate—18%/linalool—14%), *Citrus* x *sinensis* (limonene—69–91%), *Cupressus sempervirens* (α-pinene—17%/Δ-3-carene—12%), *Cymbopogon citratus* (geranial—34–45%/neral—28–31%), *C. nardus* (citronellal—39%), *Elettaria cardamomum* (eucalyptol—30%/α-terpineol acetate—28%), *Eucalyptus* sp. (eucalyptol—67%), *E. citriodora* (citronellal—40%), *Foeniculum vulgare* (anethole), *Hypericum mexicanum* (nonane—53%/α-pinene—25%), *Illicium verum* (anethole—93%), *Lavandula stoechas* (fenchone—28%/camphor—28%), *L. angustifolia* (eucalyptol—72%), *Lepechinia betonicifolia* (limonene—28%/α-pinene—19%), *Lippia alba* (carvone—35–46%/limonene—20–53%), *L. origanoides* (thymol—30–52%/α-phellandrene—25%/p-cymene—12–18%), *Minthostachys septentrionalis* (pulegone—42%), *Ocimum basilicum* (estragol—22–82%), *Ocotea* sp. (α-terpineol—44%/α-pinene—24%), *Piper* sp. (α-gurjunene—25%/elemol—14%), *P. aduncum* (dillapiole—48%/(piperitone—46%)/eucalyptol—11%/(linalool—22%)), *P. asperiusculum* (myristicin—38%/dillapiole—35%), *P. dilatatum* (apiol—89%), *P. divaricatum* (eugenol—38%/methyl eugenol—36%), *P. el-metanum* (α-pellandrene—44%), *P. gorgonillense* (β-caryophyllene—29%/α-copaene—14%), *P. nigrum* (β-caryophyllene—24%/limonene—15%), *P. pertomentellum* (*cis*-β-ocimene—28%/germacrene D—27%/trans-β-ocimene—21%), *P. sanctifelicis* (δ-3-carene—35%/limonene—27%), *Rosmarinus officinalis* (α-pinene—15–23%/eucalyptol—9–23%/(camphor—12–13%)), *Satureja viminea* (*p*-menth-3-en-8-ol—45%/pulegone—39%), *Tagetes lucida* (estragol—92%), *Xilopia discreta* (β-pinene—36%/α-pinene—25%), and *Zanthoxylum monophyllum* (β-pinene—35%/linalool—11%), whose fumigant/repellency values were as follows: on *T. castaneum*—LC_50_: 16–31 µL/L (24 h), RC_50_: 0.0005–0.05 µL/cm^2^ (2–4 h), and %Inh.: 51–100% (2–4 h) at 0.01–1 µL/cm^2^, and on *S. zeamais*—RC_50_: 0.03–0.17 µL/cm^2^ (2 h) and %Inh.: 40–97% (2–24 h) at 6–23 µL/L. The primary type of compounds that constituted the EOs from those plants was monoterpenoids, followed by phenylpropa(e)noids and sesquiterpenes.

Based on the results of repellent effects (at 2–4 h) and AChE inhibitions of the 11 EOs along with the ANOVA and multivariate analysis, five EOs were the most effective according to the (i) degree of repellency ≥ C^+^, (ii) IC_50_ values on the AChE >>> C^+^, and (iii) I% values (1 ppm) <<< C^+^. Meanwhile, other EOs, 1bL** (*P. holtonii*), 1aL (*P. holtonii*), 3bL (*P. haugtii*), 4bL^†^ (*P. reticulatum*), and 6aL (*B. simaruba*), were less active than chlorpyrifos (C^+^) in all tests. Thus, EOs 4bL^†^ (fresh leaves of *P. reticulatum*) and 6aL (*B. simaruba*—leaves) showed a time-dependent decrease in the degree of repellency (almost by half). The low repellent effect of these EOs could be attributed to the lower content (relative amount < 25% of the total) of active constituents, as well as to the molecular ratios [105], considering that β-pinene, caryophyllene oxide, and spathulenol have significant properties on Coleoptera as a fumigant (IC_50_ of 15 µg/mL) [106], toxicant (LC_50_ of 9–26 µg/insect) [107], and contact toxicant/repellent (LD_50_ of 18 µg/adult—45% mortality at 10%; 54–100% repellency at 3–79 nL/cm^2^ (2–4 h)) [108], correspondingly; meanwhile, the decrease in the repellent effect (negative chemotaxis) as a function of time could be related both to the evaporation/diffusion processes of the most volatile constituents (higher vapor pressures) and the adaptability of insects to those constituents [109,110].

Taking into consideration the other less active EOs from *P. haugtii* (3bL), *P. holtonii* (1aL), and *P. holtonii* (1bL**), they contained dillapiole as the main constituent (ca. 48–78%), which is a powerful insecticidal agent (alone or as a co-adjuvant/synergist) or a suitable repellent/fumigant compound (against *Plodia interpunctella* and *S. zeamais*, respectively) [111,112,113,114]; however, in the reviewed literature, any report on the repellent capacity of dillapiole on *S. zeamais* was not found. Furthermore, some EOs containing dillapiole as the main component showed a low/moderate repellent effect [98], which would be consistent with this report in that the three EOs were not wholly effective (lower %repellency—20–48%) against *S. zeamais* compared to C^+^. According to Fazolin et al. [111], the most potent synergistic effect of dillapiole could be verified when it was mixed with β-caryophyllene, methyl eugenol, or α-humulene; however, none of the EOs under study presented any of the mixtures of constituents mentioned above. Of course, the degree of repellency of these EOs was dillapiole-content-dependent, as seen in Figure 6, i.e., the higher the dillapiole content, the higher the degree of repellency (including EO 1bL*), which would follow what was reported by Fazolin et al. [115], who stated that the higher the dillapiole content in the EO, the higher the insecticidal effect via residual or topical contact (e.g., on *Spodoptera frugiperda*). Notably, one of the most active EOs as a repellent in this research was *P. holtonii* (1bL*), which had the highest content of dillapiole (81%).

In addition, two tests were conducted to establish the correlation between the variables % dillapiole content, % repellency, and/or IC_50_ values. Therefore, the values of Spearman´s correlation coefficients were 0.87 and −0.85 for % dillapiole content vs. %R and % dillapiole content vs. IC_50_, respectively, which showed, in turn, high positive and negative correlations. Furthermore, the hypothesis test related to the coefficients yielded *p* values < 0.05, which indicated that both correlation coefficients were statistically significant.

On the other hand, the repellent effect of the *P. holtonii* EO (1bL*) on *S. zeamais* was like that of the EOs from *Pep. pellucida* (2aL) and *B. simaruba*—bark (6aB), and there were no significant differences (*p* > 0.05) between them. The biological effect of *Pep. pellucida* EO could be attributed to its main constituents (carotol/dillapiole) and the potential synergism among them based on a report by Ali et al. [116], in which carotol demonstrated biting deterrent (minimum effective dose—MED of 25 µg/cm^2^)/repellent activities like DEET against *Aedes aegypti* and *Anopheles quadrimaculatus*. Likewise, spathulenol and β-selinene could be responsible for the repellent effect (in synergistic mode) of the EO from *B. simaruba*—bark because the two sesquiterpenoids presented an insecticidal effect on *Metopolophium dirhodum* (Hemiptera: Aphididae; an aphid pest in cereals) [117] and antifeedant activity (EC_50_ of 10.5 ± 0.3 μg/cm^2^) against *Spodoptera litura* (Lepidoptera: Noctuidae) larvae [118].

In contrast, the degree of repellency of EOs 4bL^‡^ (*P. reticulatum*) and 5aL (*B. graveolens*) was time-dependent with positive chemotaxis (increased effect), even surpassing C^+^ at 4 h; therefore, the main components of EO 4bL^‡^ (β-elemene/δ-cadinol) could be involved in the repellent action (synergistically) of the EO because these sesquiterpenoids showed repellency/contact toxicity/fumigant/antifeedant activities (LC_50_ >1–100 µg/adult, LD_50_ >100 µg/adult; weight decrease (4%) in the adult insect) against *S. zeamais*/*Megoura japonica*/*Plutella xylostella*/*Hylastinus obscurus* [119,120]. Meanwhile, the EO of *B. graveolens*—leaves consisted mainly of limonene/carvone; these monoterpenoids were effective fumigants/repellents against *S. oryzae* (100% mortality at 50 µg/mL—14 h; LC_50_ of 19 µg/mL), *Tribolium confusum* (%R of 16 ± 4%–88 ± 5%, at 0.5–64 mg), *T. castaneum* (%R > 40–90%, at 0.13–77 nL/cm^2^ (2–4 h); 100% mortality at 50 µg/mL—14 h; LC_50_ of 6 µg/mL), *Liposcelis bostrychophila* (%R 20—>90%, at 2.5–63 nL/cm^2^ (2–4 h)), *Lasioderma serricorne* (%R of 74%–80%, at 79 nL/cm^2^ (2–4 h); LC_50_ of 14 µg/mL), and *S. zeamais* (RD_50_ of 4 µg/cm^2^, LC_50_ of 52 µg/mL), which would allow us to infer their contribution to the high degree of repellency for this EO [121,122,123,124,125,126]. The comparison of the repellency by the EO from *B. graveolens*—L with other studies showed some similarities/differences; that is, Fernández-Ruiz et al. [97] found that the EO (0.02/0.2 µL/cm^2^, from Cartagena, Colombia) decreased the degree of repellency as a function of time (2 h/4 h—48 ± 10%–19 ± 15%/73 ± 8%–37 ± 19%) on *T. castaneum*. Unfortunately, the authors did not report the chemical composition of the EO. At the same time, Jaramillo et al. [61] stated that the EO (consisting of limonene (42 ± 2%)/pulegone (21 ± 1%)) from Cartagena, Colombia, showed repellent/fumigant actions on *T. castaneum*, with a repellency of 88–89% (1% EO at 2–4 h) and an LC_50_ value of 108 µg/mL (fumigant). Another manuscript related to *B. graveolens* EO was authored by Jumbo et al. [127]; the authors reported the repellency/fumigant activities of fruit EO (constituted by limonene (44%)/α-phellandrene (20%)) on *Acanthoscelides obtectus* (repellency >10–40%, 44–145 µL/L; LC_50_ of 69 µL/L) and *Zabrotes subfasciatus* (repellency >50%, 156 µL/L; LC_50_ of 71 µL/L).

Finally, the repellent effect on *S. zeamais* of the EO from *B. graveolens*—B (5aB) was time-dependent with negative chemotaxis, but at 4 h, there was no significant differences (*p* > 0.05) compared to C^+^; the main constituents of the Colombian bark EO were mintlactone (14%) and its derivative (14%), which could have a positive effect on the degree of repellency of the EO, because an extract containing mintlactone was repellent on *A. aegypti* [128]. As an entry into the discussion, it could be hypothesized that since the biogenetic precursor of mintlactone is pulegone, the derived molecule (structurally similar) could also be a powerful insecticidal agent [126,129,130].

As a last point, the data on the inhibition of AChE by EOs will be discussed with the available scientific literature; nonetheless, the *in vitro* activities on AChE of the EOs from *P. holtonii* (1bL**, 1aL, 1bL*), *Pep. pellucida* (2aL), *P. haugtii* (3bL), *P. reticulatum* (4bL^†^, 4bL^‡^), *B. graveolens*—L (5aL), and *B. simaruba* (6aB, 6aL) are reported for the first time. Thus, the IC_50_ values for *P. holtonii* and *P. haugtii* were 28 ± 2–40 ± 3 µg/mL, with *P. holtonii* (1bL*, the same EO with the best repellent effect on *S. zeamais*) being the most active (<IC_50_ value) between them. Moreover, EOs from *P. reticulatum*, *B. graveolens*, *Pep. pellucida*, and *B. simaruba* had IC_50_ values of 21 ± 1–25 ± 2 µg/mL, 16 ± 1–18 ± 1 µg/mL, 14 ± 1 µg/mL, and 4.6 ± 0.3–7.7 ± 0.9 µg/mL, respectively. The comparison between all EOs and chlorpyrifos showed that C^+^ was more active than EOs, ca. 8- to 60-fold, indicating that EOs had a lower toxicity. Nevertheless, this does not mean that EOs cannot be effective biopesticides; an example is those EOs with high amounts of dillapiole, which has proven its effectiveness as an insecticide [115,131]. In this work, it was possible to verify (Figure 6) that the higher the dillapiole content in the EOs, the higher the inhibition of the AChE enzyme (<IC_50_ values).

In the case of *P. reticulatum*, the inhibitory effect on the AChE enzyme by the two EOs was close to each other (similar IC_50_ values), and it could be related to the presence of β-pinene and β-elemene; these terpenes were insecticidal agents against *Spodoptera frugiperda* larvae (instar 2)/*Musca domestica* and *M. japonica*/*P. xylostella* with LC_50_ values of 14 µg.L^−1^ air/6 mg.dm^−3^ and >1–100 µg/adult, respectively [119,132,133]. In addition, the IC_50_ values for *B. graveolens* EOs were close, but 5aL (leaves) was slightly more active than 5aB (bark). The anti-AChE action of leaf EO could be due to monoterpenoids limonene (17%)/carvone (10%), which were effective insecticides against *S. oryzae* (LC_50_ of 19 µg/mL), *T. confusum* (LD_50_ of 33–66 µg/mL), *T. castaneum* (LC_50_ of 6 µg·mL^−1^/2–19 µg·mL^−1^, LD_50_ of 3–20 µg·insect^−1^/14 µg·insect^−1^), and *S. zeamais* (LC_50_ of 52 µg·mL^−1^/3–48 µg·mL^−1^/LD_50_ of 3–30 µg·insect^−1^/23 µg·mL^−1^/10–30 µL (90–100% mortality, 24 h)) [121,122,123,124,125,126,134,135]. In contrast, the inhibitory effect on AChE by the Colombian bark EO (composed of mintlactone/derivative) differed (being, ca. two-fold, the most active) from that reported for the Ecuadorian trunk EO (consisting of limonene (68.52 ± 0.08%)/mintlactone (20.37 ± 0.09%)), which presented IC_50_ values of 47 µg/mL and 52 µg/mL on AChE and BuChE, respectively [136]. The difference in the inhibition could be attributed to the chemical nature of constituents, i.e., mintlactone and its derivative are esters but cyclic.

On the other hand, the three most active EOs (<IC_50_ values) on the inhibition of the AChE enzyme were those from *Pep. pellucida* and *B. simaruba*. The constituents of the *Pep. pellucida* EO were carotol (44%)/dillapiole (21%), which could be responsible for the AChE inhibition (by synergy) because sesquiterpene alcohol demonstrated a high insecticidal effect (91 ± 8% of mortality at 50 µg/mL) against *A. albopictus* larvae [137]. As for dillapiole, its insecticidal capability was previously discussed. In addition, the inhibitory effect on AChE by the EO from *B. simaruba*—B could be related to the fact that spathulenol (the main component of EO) had an LC_50_ (LC_90_) value of 4.3 ± 0.2 (7.5 ± 0.8) mL/L against *M. dirhodum* (Hemiptera: Aphididae) [117]. Finally, the EO from *B. simaruba*—L (6B) was the only EO with the percentage of inhibition calculated/determined (at 1 µg/mL), as well as the one with the lowest IC_50_ value on AChE, which could be attributed to the content of caryophyllene oxide and spathulenol (in synergistic action), because they were effective insecticidal agents as described by Bettarini et al. [138], Liu et al. [139], and Kim et al. [92].

In perspective, the permanent search for new/novel active chemical agents against stored-product pests, with equal/greater effectiveness than existing pesticides but with low or no toxicity toward the consumer (humans/animals), has allowed the exploration of natural products (mainly plants) as an alternative, from which conclusive results have been found to treat these pests [140,141,142,143]. Among them, the most promising are essential oils because, in addition to being environmentally friendly, they are widely available, and there is an appropriate cost–effectiveness relationship [144]. Based on the results of this study, the five promising EOs’ varied chemical composition (phenylpropenoids, sesquiterpenoids, and monoterpenoids), high repellent effect (≥C^+^)/high IC_50_ values (on AChE) (≥C^+^)/low percentage of inhibition on AChE (≤C^+^), and fulfillment of the effectiveness/toxicity (safety) criteria (in vitro) would allow them to be included as new/novel biorepellents against *S. zeamais*. It is noteworthy that, although the EOs in this work could show a greater or lesser repellent effect compared to other EOs in the reviewed literature [67,69,71,74,82,84,85,86,88,92,97,98,100,101,102,103,104], the determination of *in vitro* toxicity (low to none) related to the AChE enzyme, which contributes to the “safety” and possible real application, would be advantageous because most of the works only evaluated the repellent effect and few evaluated the safety/toxicity criterion.

Prospectively, it is recommended to test *in vitro* the group of promising EOs against other stored-product pests (e.g., *T. castaneum*, *T. confusum*, *S. oryzae*, *Tenebrio molitor*, *Plodia interpunctella*, *Sitotroga cerealella*, *Trogoderma granarium*, *Acanthoscelides* spp., *Callosobruchus* spp., etc.), as well as evaluate them in field/storage/in situ phases (on *S. zeamais*) to establish the effective doses, and if necessary, prepare them in micro-/nano-emulsion formulations or other pharmaceutical forms to improve/potentiate the biological actions of interest.

## 4. Materials and Methods

### 4.1. Reagents and Standards

The analytical reagents used were dichloromethane (ACS grade, Alfa Aesar, Ward Hill, MA, USA), acetone (AR/GR grade, Merck, Rahway, NJ, USA), dimethyl sulfoxide (LR grade, Merck), type I water (milli-Q® Integral, Merck, Billerica, MA, USA), NaCl (≥99.5%, Merck), NaH_2_PO_4_ (99–102%, Merck), K_2_HPO_4_ (≥99%, Merck), Na_2_HPO_4_ (≥99.5%, Merck), tween^®^ 20 (polysorbate 20, Sigma-Aldrich, St. Louis, MO, USA), DTNB (5,5′-dithiobis(2-nitrobenzoic acid) ≥98%, Sigma-Aldrich), ATChI (acetylthiocholine iodide ≥98%, Sigma-Aldrich), and acetylcholinesterase enzyme (AChE) from *Electrophorus electricus* (1000 U/mg, Sigma-Aldrich).

### 4.2. Plant Materials

Samples of different parts (fresh (some dried) leaves and/or barks) of six plants (*Bursera graveolens*, *B. simaruba*, *Peperomia pellucida*, *Piper haugtii*, *P. holtonii*, and *P. reticulatum*) collected in different locations in the Departamentos de Atlántico/Sucre (Caribbean Region, Colombia) were taxonomically identified by the Instituto de Ciencias Naturales at the Universidad Nacional de Colombia. The plant collections were carried out under Resolution No. 739 of 8 July 2014, conferred by the Agencia Nacional de Licencias Ambientales (ANLA).

### 4.3. Isolation of Essential Oils and GC-MS Analysis

The EOs were isolated from fresh (or dried) parts of the plants (200–300 g) through hydrodistillation (with a modified Clevenger-type apparatus with Dean–Stark reservoir) assisted by microwave heating (Whirpool^®^ (Benton Harbor, MI, USA), domestic oven model WMS07ZDHS, operated at 700 W) for 1 hour in four 15 min cycles. Once the EOs were obtained, they were decanted, dehydrated with anhydrous sodium sulfate, and analyzed using GC-MS [145]. The chemical analysis of the EOs was then carried out using a Trace 1310 GC coupled to an ISQ Series MS (Thermo Fisher Scientific, Waltham, MA, USA), with a split/splitless inlet (split ratio of 10:1) and a liquid autosampler (AI/AS 1310 Series, Thermo Fisher Scientific). Moreover, the Rxi^®^-1ms column (30 m × 0.25 mm ID × 0.5 µm df, Restek Co., Centre County, PA, USA) was suitable for separation by individual constituents. The temperature programming of the GC oven was executed according to Muñoz Acevedo et al. [145]. Chromatographic/mass spectra data were processed/analyzed using Thermo Xcalibur^TM^ (Version 2.2 SP1.48) along with AMDIS (Build 130.53, Version 2.70) software.

Linear retention indices were calculated using a C_7_-C_35_ linear hydrocarbon mixture and analyzed under the same conditions as the samples. The chemical components were identified by comparing their mass spectra and linear retention indices with those of available databases (NIST11, NIST Retention Index, and Wiley9) and the consulted/existing literature [146,147,148,149,150].

### 4.4. NMR Analysis

Hydrogen (^1^H) and carbon (^13^C) NMR spectra were acquired at 400 MHz and 100 MHz, respectively, on an Avance-400 Bruker spectrometer. Chemical shifts were reported in ppm using TMS as an internal reference (δ scale), and CDCl_3_ was used as a solvent and an internal standard (^1^H: δ 7.26 ppm; ^13^C: δ 77.00 ppm).

### 4.5. Collection and Breeding of Maize Weevils

Coleoptera (*Sitophilus zeamais* Motschulsky) were collected from infested corn purchased in commercial grain stores in Barranquilla (Colombia). Then, they were grown in a controlled environment using maize grains in good conditions, with 70% humidity at 25 °C for six weeks, as described by Throne [151]. As soon as a representative number of weevils reached the adult stage, the assay was carried out along with their replicates.

### 4.6. Implementation of Repellency Test

The degree of repellency on maize weevils by the samples (EOs/chlorpyrifos (positive control)) was evaluated by applying the contact method through paper impregnation (preferred area (Tapondjou et al. [152])) fitted to a modified tunnel-type device (Figure 7—all parts of this device were transparent and odorless). In this method, a disk (ø 5.2 cm) of filter paper along with one previously cleaned healthy maize grain was placed inside each polypropylene Petri dish (V: 26.6 cm^3^ (ø: 5.2 cm, h: 1.25 cm), compartments 1 and 2). The pieces (filter paper and corn grain) in the first compartment were impregnated with 26.55 µg of each sample (EOs/control), formerly dissolved up to 300 μL with acetone. Once the solvent was evaporated (ca. 1.25 µg/cm^2^ or 1 µg/cm^3^ air was the sample concentration in the dish), ten adult maize weevils (unsexed and uneaten for 24 h) were placed inside the compartment, which was subsequently sealed. From this moment, the experiment started and was monitored at 2 h and 4 h.

During observation, the effect was considered positive/measurable when maize weevils moved from treated to untreated areas. The degree of repellency (in percentage value—%R) was calculated based on Equation (1).
(1)$$\%R=\frac{I_{UTZ}}{I_{UTZ}+I_{TZ}}*100,$$
where I_UTZ_ and I_TZ_ are the number of individuals found/counted in the untreated and treated areas, respectively. All experiments were performed in quintuplicate, with their positive/negative controls and the respective statistical treatment of the data using IBM SPSS Statistic 27 software.

### 4.7. Acetylcholinesterase Inhibition Assay

The *in vitro* inhibitory effect of EOs/chlorpyrifos on the AChE enzyme was measured in agreement with the colorimetric method reported by Ellman et al. [153], for which the samples (EOs/chlorpyrifos, 50 µL of each), prepared at 4–125 µg/mL (0.3–4.8 µg/mL—chlorpyrifos) and 1 µg/mL (all samples), were placed to react in a 96-well plate with AChE (50 µL—0.25 U/mL) for 30 min at 25 °C (continuous shaking). Afterward, the substrate (100 µL—consisting of DTNB (0.2 mM), AChI (0.24 mM), and Na_2_HPO_4_ (0.04 M)) was added to each well, and the final total mixture was incubated at 37 °C during six minutes and analyzed in a 96-well plate reader at 412 nm. All solutions of the samples and enzyme were prepared in a PBS buffer (pH 7.5). The enzymatic activity was observed when the yellow color increased via the formation of the thionitrobenzoate anion through the reaction between the dithiobisnitrobenzoate anion and thiocholine. The percentage of enzyme inhibition (%I) was calculated according to Equation (2).
(2)$$\%I_{\lambda -412}=100-\left[\frac{\left(AS-AB\right)}{\left(AC-AB\right)}\right]*100,$$
where AS, AC, and AB are the absorbances measured at six minutes of the potential inhibitor (samples—EOs/chlorpyrifos), control, and blank, respectively. The IC_50_ (50% inhibitory concentration) values were obtained from the graphs of the percentage of inhibition (at six minutes) versus the concentration of the evaluated substance. All experiments were performed in quintuplicate, with their positive/negative controls and the respective statistical treatment of the data using IBM SPSS Statistic 27 software.

### 4.8. Statistical Analysis

The raw data of the repellency results were treated using a two-way (exposure time and type of EO) analysis of variance (ANOVA, *p* < 0.05) and one-way (type of EO) ANOVA combined with straight line regression, along with the following tests: Tukey HSD (comparing the effects among treatments (*p* < 0.05)), Dunnett (comparing the effects between each treatment and single control (*p* < 0.05)), Kolmogorov–Smirnov (verifying the assumption of normality of the errors (*p* > 0.05)), and Levene (proving the assumption of homoscedasticity (*p* > 0.05)). In addition, the Spearman correlation test (*p* < 0.05) was applied to establish correlations between dillapiole content and %R and IC_50_ values. All acquired data on the degree of repellency and inhibitory effect of the samples were statistically treated and subjected to PCA, CA, and KmCA as tools of a multivariate statistical analysis using IBM SPSS Statistic 27 (2020), Statgraphics 18 (2020), and R core 4.0.3 (2020) software.

## 5. Conclusions

Five essential oils (*P. holtonii*—leaves (1bL*), *Pep. pellucida*—leaves, *B. simaruba*—bark, *B. graveolens*—bark, and *B. graveolens*—leaves) from the northern region of Colombia were promising based on the repellent capacity on *S. zeamais* and *in vitro* inhibition of the AChE enzyme. They were mainly constituted by dillapiole, carotol, spathulenol, limonene, and mintlactone, which could be responsible for the bioproperties of the EOs. In addition, these EOs could be used as protective agents against attacks by Coleoptera insects on stored products (e.g., maize), exerting an effective repellency at a relatively low concentration, possibly with low toxicity (total/residual) in humans.

## Figures and Tables

**Figure 1 molecules-29-01753-f001:**
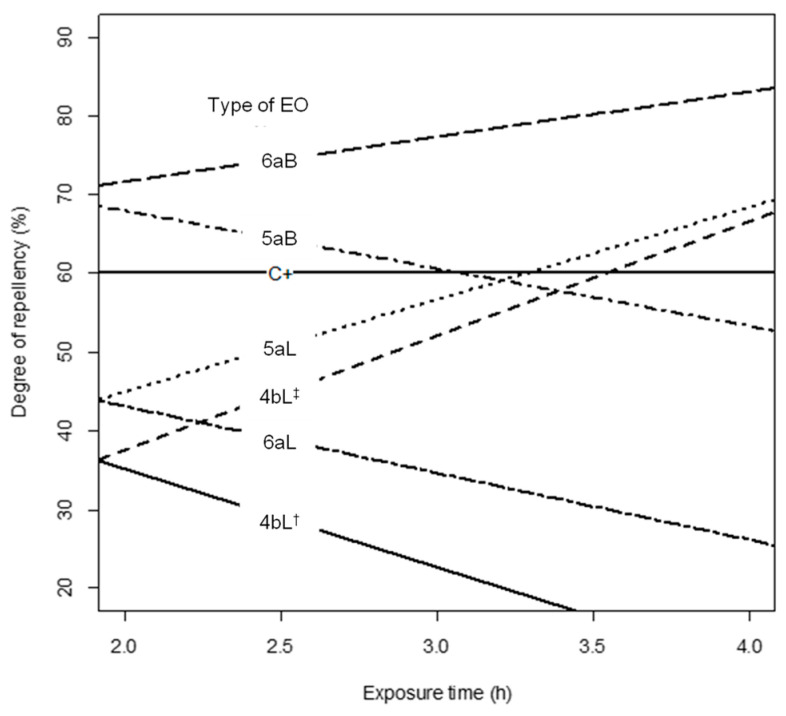
Comparison of the repellent effect against *S. zeamais* between C^+^ (chlorpyrifos) and the 11 EOs according to the exposure time (2 h and 4 h), based on model (2).

**Figure 2 molecules-29-01753-f002:**
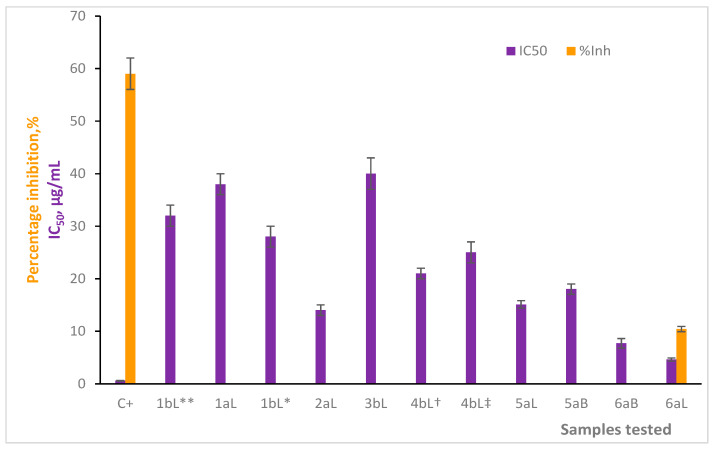
Inhibitory effects (IC_50_ and %I (at 1 µg/mL)) of the 11 EOs tested and chlorpyrifos on AChE.

**Figure 3 molecules-29-01753-f003:**
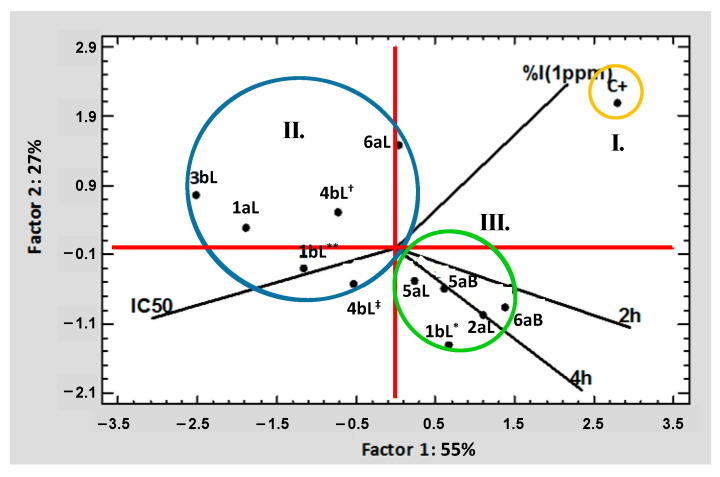
PCA includes the values of IC_50_, %I (at 1 µg/mL), and degree of repellency of EOs and chlorpyrifos.

**Figure 4 molecules-29-01753-f004:**
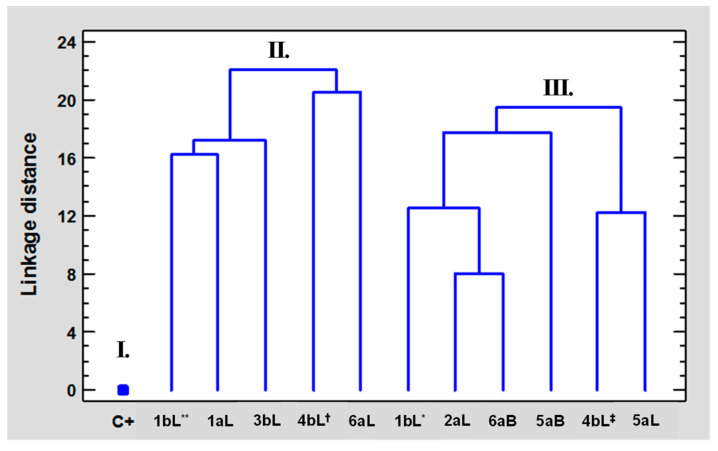
Vertical hierarchical tree plot from the CA related to the 11 EOs and chlorpyrifos, based on the values of IC_50_, %I (at 1 µg/mL), and degree of repellency.

**Figure 5 molecules-29-01753-f005:**
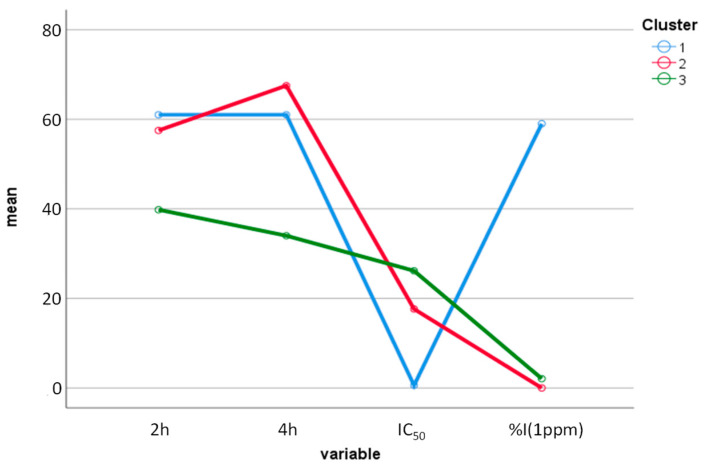
Plot of the means for the three clusters based on the values of IC_50_, %I (at 1 µg/mL), and repellency.

**Figure 6 molecules-29-01753-f006:**
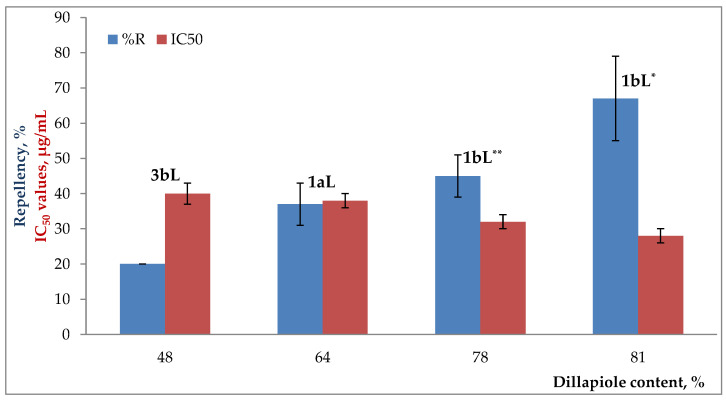
Effect of dillapiole content from EOs on the degree of repellency against *S. zeamais* (blue bars) and IC_50_ values on AChE (red bars).

**Figure 7 molecules-29-01753-f007:**
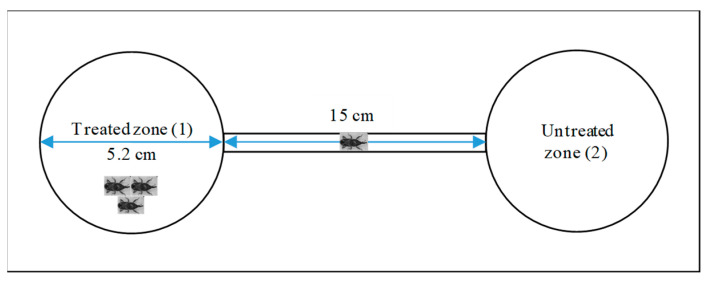
Modified tunnel-type device to measure the repellent effect on maize weevils (sketch made by authors).

**Table 1 molecules-29-01753-t001:** Main constituents identified in the 11 EOs from six plants.

Constituents ^Ϯ^	R_I_		Relative Amount, %										
	Cal.	Lit.	1bL**	1aL	1bL*	2aL	3bL	4bL^†^	4bL^‡^	5aL	5aB	6aB	6aL
β-Pinene	969	970	---	---	---	---	---	8.2	---	---	---	---	---
*p*-Cymene	1010	1011	---	---	---	---	---	---	---	---	---	---	5.0
Limonene	1019	1020	---	---	---	---	---	---	---	16.6	---	---	---
*E*-β-Ocimene	1039	1036	---	---	---	---	4.0	---	---	---	---	---	---
Linalool	1081	1081	---	---	---	---	---	6.9	---	---	---	---	---
Terpinen-4-ol	1154	1160	---	---	---	---	---	---	---	---	---	---	6.1
trans-Carveol	1196	1195	---	---	---	---	---	---	---	5.4	---	---	---
Carvone	1208	1213	---	---	---	---	---	---	---	10.0	5.6	---	---
Piperitone	1218	1228	---	---	---	---	6.2	---	---	---	---	---	---
Pulegone	1223	1237	---	---	---	---	---	---	---	---	8.4	---	---
Limonene-1,2-diol	1310	1321	---	---	---	---	---	---	---	7.7	6.2	---	---
Mintlactone derivative	1322	----	---	---	---	---	---	---	---	---	14.2	---	---
β-Bourbonene	1374	1386	---	---	---	---	---	---	---	5.7	---	---	---
β-Elemene	1380	1387	---	---	---	4.7		7.9	8.6	---	---	---	---
β-Caryophyllene	1407	1418	---	---	---	7.3	4.8	6.7	4.4	---	---	---	---
Mintlactone	1461	1472	---	---	---	---	---	---	---	---	13.5	---	---
Germacrene D	1465	1470	5.2	5.2	3.7	---	---	7.8	---	---	---	---	---
β-Selinene	1473	1483	---	---	---	---	---	---	---	---	---	17.6	---
Bicyclogermacrene	1490	1487	3.4	6.3	5.7	---	---	---	---	---	---	---	---
Caryophyllene oxide	1551	1558	---	---	---	---	---	---	---	---	---	9.9	12.0
Spathulenol	1578	1577	---	---	---	---	---	---	4.3	---	---	24.3	10.9
Carotol	1579	1590	---	---	---	43.7	---	---	---	---	---	---	---
Dillapiole	1580	1593	78.5	64.4	80.9	20.9	48.2	---	---	---	---	---	---
δ-Cadinol	1633	1646	---	---	---	---	---	---	5.1	---	---	---	---
α-Cadinol	1638	1641	---	---	---	---	---	---	4.2	---	---	---	---

1bL**—*P. holtonii*, 1aL—*P. holtonii*, 1bL*—*P. holtonii*, 2aL—*Pep. pellucida*, 3bL—*P. haugtii*, 4bL^†^—*P. reticulatum*, 4bL^‡^—*P. reticulatum*, 5aL—*B. graveolens*, 5aB—*B. graveolens*, 6aB—*B. simaruba*, 6aL—*B. simaruba*, R_I_—Retention indices on column Rxi-1ms, Calc—Calculated, Lit—Literature, L—Leaves, B—Bark, a—Departamento del Atlántico, b—Departamento de Sucre, * Location I, ** Location II, ^†^ Fresh, ^‡^ Dried, ^Ϯ^ Compounds identified based on mass spectra (obtained using GC-MS/compared to spectral libraries) and linear retention indices (calculated/from the scientific literature).

**Table 2 molecules-29-01753-t002:** Repellent effect (at 2 h and 4 h) against *S. zeamais* of the 11 EOs and chlorpyrifos (control substance).

Code	Sample Tested	Degree of Repellency (%) ^Ϯ^	
		2 h	4 h
C^+^	Chlorpyrifos	61 ± 8	61 ± 8
^¥^ 1bL**	*P. holtonii*	45 ± 6	48 ± 5
^¥^ 1aL	*P. holtonii*	37 ± 6	34 ± 6
1bL*	*P. holtonii*	67 ± 12	73 ± 6
2aL	*Pep. pellucida*	65 ± 6	70 ± 10
^¥^ 3bL	*P. haugtii*	20 ± 0	32 ± 5
^¥^ 4bL^†^	*P. reticulatum*	52 ± 5	28 ± 5
^¥^ 4bL^‡^	*P. reticulatum*	38 ± 5	67 ± 6
5aL	*B. graveolens*	45 ± 6	68 ± 12
5aB	*B. graveolens*	68 ± 8	53 ± 8
6aB	*B. simaruba*	62 ± 5	74 ± 11
^¥^ 6aL	*B. simaruba*	45 ± 6	28 ± 4

L—Leaves, B—Bark, a—Atlántico, b—Sucre, * Location I, ** Location II, ^†^ Fresh, ^‡^ Dried, ^Ϯ^ Values reported as $\bar{X}$ ± s according to the replicates. ^¥^ Statistically significant differences (*p* < 0.05) with respect to C^+^. Codes for plants are numbers: different types of species (1—*P. holtonii*, 2—*Pep. pellucida*, 3—*P. haugtii*, 4—*P. reticulatum*, 5—*B. graveolens*, 6—*B. simaruba*).

## Data Availability

The most important data were included both on the manuscript and Appendix A.

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
