# Peer review of "The Repellent Capacity against Sitophilus zeamais (Coleoptera: Curculionidae) and In Vitro Inhibition of the Acetylcholinesterase Enzyme of 11 Essential Oils from Six Plants of the Caribbean Region of Colombia"

_molecules, 2024, doi:10.3390/molecules29081753_

Round 1

Reviewer 1 Report

Comments and Suggestions for Authors

In this paper, the repellent capacity against Sitopilus zeamais 11 essential oils (EOs) isolated of six plants were measured, and their components of EOs and its inhibition on AChE were analyzed. It show us a  valubale results and wrote with smoth english but the manuscript could be improved more in some items. 

Some comments:

1. how to identify the plants? and this may be need not to show this paper

2. how to identify the main constituents in EOs, do you have a standard

3. the same result presented with two manner, please select only table or figure

4. the significant difference need to show in figure or table

5. all the main constituents analyzed by the NMR?

6. the model of statistical treatment (Line 147-178) is not necessary, or it could be describe in method simply.

7. it is recommended to use the correlation test to analyze the repellency capacity and IC50, and dillapiole

8. this representation "20 ± 0-68 ± 8%" is not very good for reading

9. line 159 (p 0.075)

Comments on the Quality of English Language

the manuscript wrote with well English.

Author Response

Thank you very much to the Reviewer 1 for all the suggestions/comments. All of them were taken into account to improve the presentation of the manuscript. Answers to specific questions/suggestions are given below:

*Does the introduction provide sufficient background and include all relevant references? (can be improved)

Thank you very much for the suggestion… The introduction was improved, as well as all relevant references were included.

*Are all the cited references relevant to the research? (can be improved)

All cited references relevant were included.

  1. how to identify the plants? and this may be need not to show this paper

Thank you very much to the reviewer for the question and suggestion… The plants under study were identified by botanists (experts in specific families) from the Herbario Nacional de Colombia at the Instituto de Ciencias Naturales of the Universidad Nacional de Colombia (It is mentioned in materials and methods, item plant material). The plants were identified using taxonomic criteria, and one specimen of each plant (witness sheet) was permanently deposited in the Herbarium, which was assigned a consecutive identification code (COL No.). According to the instructions for authors of Molecules, “For each submitted manuscript, supporting genetic information and origin must be provided. For research manuscripts, voucher specimens must be deposited in an accessible herbarium or museum…” Thus, for each plant, the COL number was included in the manuscript in the section Results …

  1. how to identify the main constituents in EOs, do you have a standard

Thank you very much to the reviewer for the question. The main constituents of the EOs were identified as a first approximation using chromatographic (tR and calculated retention indices) and spectrometric (mass spectra) criteria. Those data were compared with the available scientific literature related to retention indices [NIST Retention Index database, the NIST web page (webbook of chemistry - Linstrom and Mallard, 2022) and the articles of Davies (1990) and Babushok et al. (2011)] and mass spectra (NIST11, NIST Retention Index, and Wiley9 databases, Adams 2017/Joulian et al. 1998 - books). In addition, some main constituents of the Colombian essential oils (studied by our research team) have been chemically characterized by 1H-NMR (or 13C-NMR) (methyleugenol, germacrene D, dillapiole, myrcene, alpha-/beta-eudesmol, bycyclogermacrene, cis-piperitone oxide, limonene, gibbilimbol B) that along with the chromatographic/spectrometric (tR/calculated retention indices/mass spectra) criteria mentioned above have guaranteed an unequivocal structural elucidation allowing the construction of an own database of the chemistry of Colombian EOs. Of course, the lab counts with some certified standards of constituents of EOs, but they were not used because they were absent in the EOs under study. Again, it is mentioned that in this study, the most important (most active) essential oil contained dillapiole. Then, the EO with the highest dillapiole content was analyzed by 1H-/13C-NMR, to confirm the structure and validate the structural information obtained by GC-MS, supporting the previously mentioned chromatographic and spectroscopic criteria.

  1. the same result is presented with two manners, please select only table or figure

Thank you very much for the suggestion… Only one form of results was included (Table)

  1. the significant difference need to show in figure or table

Thank you very much for the suggestion… Done (Table)

  1. all the main constituents analyzed by the NMR?

Thank you very much to the reviewer for the question. This analysis was not performed on individual constituents. The NMR analysis was applied only for the EO with the highest content of dillapiole (81%) as a whole because the higher the content of a component, the greater the NMR signals, characterizing the main constituent, giving identity to this EO.

  1. the model of statistical treatment (Line 147-178) is not necessary, or it could be describe in method simply.

Thank you very much for the suggestion… Done. The equation models were removed and transferred to supplementary materials. Only a quick description of each model was included.

  1. it is recommended to use the correlation test to analyze the repellency capacity and IC50, and dillapiole

Thank you very much for the suggestion… a correlation test was applied to data between IC50, %R, and dillapiole content. The comment about that was included in the manuscript…

  1. this representation "20 ± 0-68 ± 8%" is not very good for reading

Thank you very much for the suggestion… Done

  1. line 159 (p 0.075)

Thank you very much for the suggestion… Done

Reviewer 2 Report

Comments and Suggestions for Authors

It is a pleasure to review this manuscript, which focuses on an innovative approach to bio-repellency and enzyme inhibition via essential oils derived from specific plants endemic to Colombia's northern region. The study's pertinence is rooted in the growing necessity for environmentally friendly and sustainable alternatives to synthetic chemical agents, especially in pest control contexts, where the balance between efficacy and safety is precarious. This research's merit lies in its exploration of non-traditional, botanical solutions, specifically essential oils, and their interactions with Sitophilus zeamais, a significant pest affecting stored products, primarily grains like maize, and the in vitro acetylcholinesterase (AChE) inhibitory activity, a relevant factor in pest management. However, there are some problems to be addressed in the current version.

1.     Line 12: "11 Essential oils, isolated of six plants from the Colombia's northern region, were assessed...", the number should be written in words, and there's a grammatical error in the phrasing. Use "Eleven essential oils, isolated from six plants from Colombia's northern region, were assessed...".

2.     Line 13: "...in the repellent capacity against Sitophilus zeamais and the in vitro inhibition on AChE, using a type-tunnel device modified and the Ellman colorimetric method, respectively." the sentence structure is awkward. "...for their repellent capacity against Sitophilus zeamais and their in vitro inhibition of AChE, using a modified type-tunnel device and the Ellman colorimetric method, respectively." is better.

3.     "...these EO were constituted by: spathulenol...". Plural inconsistency, replace “EO” to “EOs”.

4.     As you said in the introduction, the aim of this work was to establish the degree of repellency against the maize weevil and the in vitro inhibitory capacity on AChE of the eleven EO isolated from six plants, so what is the purpose of chlorpyrifos treatment?

5.     The code in Table 2 is very confusing. Why are they code like this? I understand that you want to explain that 1, 1B, and 1C are all from Piper holtonii, but there is no any explanation in the table notes.

6.     Numerous data analysis methods are displayed in Results section 2.3, although these methods belong in Materials and Methods, results only shows the results of the experiment.

7.     What does I% in table 3 mean? It is %I in line 210 and 214, why?

8.     Is it necessary to use biometric data for PCA and cluster analysis? why? Is it meaningful?

9.     The discussion section of your paper extensively restates the data and findings presented in the results section. While it is important to refer back to key findings in the discussion, it should be more than a recapitulation of the results. The emphasis should be on interpreting the results, providing insights, drawing conclusions, and possibly guiding future research directions.

Comments on the Quality of English Language

I recommend a thorough proofreading and language polishing of the manuscript. This process should not only focus on grammatical correctness but also on the use of clear, concise, and coherent scientific communication. If necessary, seeking assistance from a professional language editing service might be highly beneficial.

Author Response

Thank you very much to Reviewer 2 for all his suggestions/comments. All of them were taken into account to improve the presentation of the manuscript. Answers to specific questions/suggestions are given below:

  1. Line 12: "11 Essential oils, isolated of six plants from the Colombia's northern region, were assessed...", the number should be written in words, and there's a grammatical error in the phrasing. Use "Eleven essential oils, isolated from six plants from Colombia's northern region, were assessed...".

Thank you very much for the suggestion… Done. The entire manuscript was revised/corrected in the English language.

  1. Line 13: "...in the repellent capacity against Sitophilus zeamaisand the in vitro inhibition on AChE, using a type-tunnel device modified and the Ellman colorimetric method, respectively." the sentence structure is awkward. "...for their repellent capacity against Sitophilus zeamais and their in vitro inhibition of AChE, using a modified type-tunnel device and the Ellman colorimetric method, respectively." is better.

Thank you very much for the suggestion… Done. The entire manuscript was revised/corrected in the English language.

  1. "...these EO were constituted by: spathulenol...". Plural inconsistency, replace “EO” to “EOs”.

Thank you very much for the suggestion… Done. The entire manuscript was revised/corrected in the English language.

  1. As you said in the introduction, the aim of this work was to establish the degree of repellency against the maize weevil and the in vitro inhibitory capacity on AChE of the eleven EO isolated from six plants, so what is the purpose of chlorpyrifos treatment?

Chlorpyrifos is a well-known, effective and common organophosphate pesticide widely used in households and agriculture (for crop protection against insects, e.g., corn – Giesey et al. 2014, Springer), which has a semi-volatile chemical nature (it has a slightly skunky odor/mild mercaptan or onions/garlic; Pvap. 0.0025 Pa; KH: 6.6x10-6 atm-m3/mol (KH < 10-5 atm-m3/mol can volatilize slowly)) and the inhibition of the AChE enzyme is the action mechanism by which this pesticide acts. Likewise, the volatilization of CP is a significant dissipative process in the environment. Moreover, it is one of the few pesticides that has shown moderate toxicity by inhalation (>0.2 mg/L in rats, between 4-6 h – National Pesticide Information Center). This pesticide was chosen as a positive control for the research due to all the characteristics described above wide use in agriculture (e.g., the crop of corn), characteristic odor (allowing its application as a fumigant – related to “odor” and “volatility”), and moderate toxicity by inhalation. In addition, the CP concentration used in the repellency test did not cause the death of maize weevils. In most cases, it resulted in repellency; in others, it led to “stunning” and “fainting.” CP was the most appropriate positive control for all the previous reasons above DEEP.

  1. The code in Table 2 is very confusing. Why are they code like this? I understand that you want to explain that 1, 1B, and 1C are all from Piper holtonii, but there is no any explanation in the table notes.

Thank you very much for the suggestion… Done. An explanation of letter/number codes used in Tables or figures was included as a note in Table 1.

  1. Numerous data analysis methods are displayed in Results section 2.3, although these methods belong in Materials and Methods, results only shows the results of the experiment.

Thank you very much for the suggestion… Done. The results in the section 2.3 were revised and adjusted.

  1. What does I% in table 3 mean? It is %I in line 210 and 214, why?

 Thank you very much for the suggestion… The change was made (%I), although such a table was transferred to supplementary materials.

  1. 8.     Is it necessary to use biometric data for PCA and cluster analysis? why?Is it meaningful?

Thank you very much for the questions. For the first and second questions, the answers are (i) biometric data could be used for PCA/CA when the main slope of analysis is for finding authentication/recognition parameters (fingerprints, facial, iris, and palm or finger vein patterns; or morphological/physiological properties) between individuals (human, plants, animal); (ii) because both techniques help make decisions, then they must be applied along with artificial intelligence or artificial neural network (which are trained/instructed) for the case of people/animal due to biometric verification is a highly complex procedure... and for the third question, the answer is yes, of course, if the scope/objective is to identify/recognize a unique signature of individuals (authentication/recognition patterns) based on physical/morphological/physiological characteristics. It is very important to make precise that biometric data were not included/used for PCA because the last purpose of this research was to correlate the chemical constituents (not just one compound) of EOs with biological activities (promising). Of course, such EOs are components of some plants (not just one plant) under study. As a complement to the answers given, the following is argued: PCA has been used to statistically treat biometric data because it is a type of algorithm in biometrics. However, biometric verification (of unique signatures of individuals) is a very complicated procedure involving pattern recognition, signal, and image processing technologies. In most cases, it is necessary to employ artificial intelligence-based approaches. Then, PCA is applied for decision-making…better recognize images – face recognition/image compression. In addition, PCA and cluster analysis are multivariate exploratory methods used to analyze data statistics (any type, e.g., multicollinearity, missing values, categorical data, and imprecise measurements), including biometric data, because the PCA is a very flexible tool and is applied in various fields, from neuroscience to financial services, finding patterns in high dimensional/complexity data. Thus, PCA identifies the main directions and strengths computing (orthogonal transformation) the covariance matrix of data to convert a set of observations of possibly correlated variables in a group of values of linearly uncorrelated variables, i.e., it reduces multidimensional data (when complex) to lower dimensions while retaining most of the information, extracting primary components, and rejecting noise. It covers standard deviation, covariance, and eigenvectors (Chen et al. 2002, IEEE ICIP; Karamizadeh et al. 2013, J. Signal Inform. Proc.; Ye et al. 2007, IEEE Xplore; Jolliffe and Cadima, 2016, Phil. Trans. R. Soc. A; Stojanović et al. 2016, Int. J. Phytoremediation; Khandelwal et al. 2016, Proc. Comp. Sci.).

  1. The discussion section of your paper extensively restates the data and findings presented in the results section. While it is important to refer back to key findings in the discussion, it should be more than a recapitulation of the results. The emphasis should be on interpreting the results, providing insights, drawing conclusions, and possibly guiding future research directions.

Thank you very much for the suggestion… Done. The discussion was revised and corrected according to the comment. However, the discussion was based on the findings of this work, which were compared with other publications related to this topic found in the scientific literature, highlighting similarities and differences between the biological effects and the possible responsibility of the constituents in such effects… in addition, the results were interpreted. Still, conclusions were included in the corresponding item (5. Conclusions – according to the Molecules template), and future research guides were also mentioned/included…  You can check it in the new version of the manuscript.

Comments on the Quality of English Language

I recommend a thorough proofreading and language polishing of the manuscript. This process should not only focus on grammatical correctness but also on the use of clear, concise, and coherent scientific communication. If necessary, seeking assistance from a professional language editing service might be highly beneficial.

Thank you very much for the suggestion… Done.

Round 2

Reviewer 2 Report

Comments and Suggestions for Authors

If you think CP is the most appropriate positive control in this study, then you should at least add an appropriate description in the introduction.

Comments on the Quality of English Language

Grammatical issues in the manuscript were improved.

Author Response

Thank you very much to reviewer 2 for your suggestion. The answer is given below...

Comments and Suggestions for Authors

If you think CP is the most appropriate positive control in this study, then you should at least add an appropriate description in the introduction.

Thank you very much for the suggestion… Done. The introduction was corrected/improved, that is, a paragraph about chlorpyrifos was included; as well as, why chlorpyrifos was chosen/used as a positive control.